REVIEW-SYMPOSIUM

# Can we study whisker movements to gain insights into the natural sensory behaviours of mammals?

Robyn A. Grant [ID]

*Department of Natural Science, Manchester Metropolitan University, Manchester*

Handling Editors: Laura Bennet & Ilan Lampl

The peer review history is available in the Supporting Information section of this article (https://doi.org/10.1113/JP288053#support-information-section).

**Abstract figure legend** The recommended future of whisker science research is integrating findings from the laboratory with studies from other captive institutions (such as zoos, rehabilitation centres and specialist research institutes) and field observations.

**Abstract** Neuroscientists, behavioural scientists, mechanical engineers and roboticists collaborate in the broad field of whisker science to investigate tactile sensing and movement in mammals. Much of this research is focussed on the study of laboratory rodents, with important insights already gained from studying their whisker movements, control behaviours and the sensory processing of whisker signals. The findings of whisker behaviour studies in the laboratory have also formed the foundation for research in other captive settings, such as in zoos. However, without inspiration from more natural environments and stimuli, researchers are probably missing out on describing other important whisker behaviours, which may in turn give researchers better insights into the brain areas, signals and behaviours associated with active whisker touch sensing. Taking inspiration from recent findings from the field and zoo, developing more social and active foraging tasks for the laboratory would probably enrich whisker behaviour studies, as would including a wider variety of species. In the longer-term, a

**Robyn Grant** is a Reader in Comparative Physiology and Behaviour at Manchester Metropolitan University, UK. She is a Sensory Biologist specialising in somatosensation and the Editor-in-chief of *Mammal Review*. Robyn's research addresses fundamental questions about the form and function of vertebrate touch sensing by adopting an interdisciplinary approach. Specifically, her work focusses on facial touch sensors: whiskers in mammals and bristles in birds.

The Journal of Physiology

more integrated approach, with collaboration across laboratory, captive and field settings, will help to develop more natural behavioural tasks representative of what an animal experiences in the real world, which would give us greater insights into the natural sensory behaviours of mammals. This has implications for the fields of neuroscience, sensory biology and evolutionary biology, as well important applications for captive mammal health and welfare.

(Received 12 March 2025; accepted after revision 13 May 2025; first published online 12 June 2025)

**Corresponding author** R. A. Grant: Department of Natural Science, Manchester Metropolitan University, Manchester, M1 5GD.    Email: robyn.grant@mmu.ac.uk

## Introduction

Whiskers, or vibrissae, are present across mammals, in most species, from marsupials to primates (Ahl, 1986; Muchlinski et al., 2020). They are specialised vibrotactile sensors, which are mainly used to guide locomotion and foraging by touch (Grant & Goss, 2022). Some species can also use their whisker follicles to sense wind (Mugnaini et al., 2023), water movements (Hanke et al., 2010; Krüger et al., 2018) and even electromagnetic fields (Czech-Damal et al., 2012). The field of whisker science is broad; from neuroscientists to comparative biologists, and from engineers to bio-inspired roboticists. In this field, scientists are working together to understand how the mammalian brain processes whisker signals, and how whiskers can be moved to maximise sensory information. To understand these questions, we are increasingly realising the importance of studying animals using more naturalistic, freely-moving, scientific protocols (Miller et al., 2022; Parker et al., 2020). Early studies in whisker research recorded from neurons in the whisker system using anaesthetised rodents, with whisker touches being introduced passively (Gibson & Welker, 1983; Zucker & Welker, 1969). However, anaesthesia disrupts neuronal signals and decreases the blood oxygen level response (Aksenov et al., 2015), and passive whisker stimulation does not give rise to the same neuronal signals as an animal actively moving their whiskers (Campagner et al., 2018). Therefore, whisker tasks should use awake, behaving animals.

Miller et al. (2022) have emphasised the importance of studying natural behaviour in neuroscience; specifically, by developing experimental protocols to match nature more closely. For example, by developing sensory stimuli representative of what an animal naturally experiences, as well as behavioural tasks that enable a broad repertoire of actions (Miller et al., 2022). Many neuroscientists are now moving beyond simplified laboratory setting into more complex and nature-inspired testing scenarios; for example, by developing novel navigation and foraging tasks (Cisek & Green, 2024). The observations from these studies reveal that much richer neuronal signals occur when an animal is freely moving and naturally behaving (Miller et al., 2022; Parker et al., 2020), supporting the need for more nature-inspired laboratory tasks. Observing awake and naturally behaving animals is particularly important when studying whiskers because they not only sense, but also move, using a complex architecture of muscles (Dörfl, 1982; Haidarliu et al., 2010). Laboratory rodents move their whiskers at ∼8–25 times per second (Mitchinson et al., 2011), termed whisking (Fig. 1). As well as whisking, studies of freely moving and naturally behaving animals have also revealed complex control behaviours that occur during head rotations (Towal & Hartmann, 2006), locomotion (Arkley et al., 2014) and object contact (Carvell & Simons, 1995; Grant et al., 2009) (Fig. 1). These behaviours serve to orient whiskers towards salient spaces in the environment and increase the efficiency of whisker sensing by controlling contact parameters, such as force, and the number of whisker touches (Grant et al., 2009). Therefore, whisker positions can be thought of as revealing a zone of attention (Arkley et al., 2014; Mitchinson & Prescott, 2013), and give us important insights into perception.

Recently, whisker behaviour tasks and findings from the laboratory have been used to inspire studies in other captive settings, such as zoos, and even into field-based settings. For example, a classic novel object task has been employed in zoos to study the whiskers of sixteen different species and this revealed that all the species moved their whiskers and engaged in control behaviours too (Grant et al., 2023). Observations from the field have even shown that elephant seals might even whisk their whiskers (Adachi et al., 2022). These studies showcase the promising future of whisker research, being broad and comparative and taking place in various captive and field-based settings. Indeed, the whisker research field is already multidisciplinary and is therefore uniquely suited to further develop working practices across lab, captive and field environments.

This review summarises whisker behaviour findings from the laboratory, zoo and field. It first provides examples where findings from the laboratory have helped to inform research studies in other captive settings, as

well as the field. The findings of these examples will then be considered, especially focussing on how observations from the field can now be used to feed into our practices in laboratory and captive settings. It is certainly an insurmountable task to move the field of whisker-based neuroscience into the wild; however, observing animals in more natural settings might serve to inspire more naturalistic tasks in the laboratory. It is only with such an integrated approach that we will be able to describe the full repertoire of natural whisker behaviours, as well as understand how such behaviours might be controlled and processed in the brain.

## Whisker studies in the laboratory

The earliest studies in whisker science were behavioural, with in-depth qualitative descriptions of rat behaviour, especially following whisker removal (Vincent, 1912, 1913). There were also many comparative studies, with a focus on whisker layouts and neuroanatomy (Ahl, 1986; Woolsey et al., 1975). However, with the improvement of neural recording techniques, the rodent whisker system became the primary model from which to understand sensory processing in the brain. There are several reviews of these findings available, including Diamond et al. (2008), Diamond (2010), Campagner et al. (2018) and Evans et al. (2019). Many studies focussed on the barrel cortex and on tracing the sensory signal from whisker touch to cortex (Petersen, 2019; Staiger & Petersen, 2021). This foundation of whisker science, being firmly embedded in laboratory neuroscience, has impacted the design of research studies. Specifically, simple stimuli, such as air puffs and movable metal poles, are now standard whisker stimulators (Bosman et al., 2010; Campagner et al., 2018), rather than the selection of a stimulus that mimics natural whisker interactions.

Behavioural studies also occurred at this time, and the whiskers were filmed and manually tracked for the first time, using tracing paper laid over the footage to precisely measure the whisker angles (Carvell & Simons, 1995; Wineski, 1983). Whiskers are inherently small, and are moved quickly, which makes them hard to see and measure. These behavioural studies provided the first quantification of the cyclic whisking that rodents make (Fig. 1). They described the frequency of whisking, which is ~8 times per second in rat (Carvell & Simons, 1995) and 16 times per second in Golden hamsters (Wineski, 1983), and suggested that both protraction and retraction stages of the whisk were under active control (Berg & Kleinfeld, 2003). Researchers also observed changes in whisking following contact (Berg & Kleinfeld, 2003; Carvell & Simons, 1995), especially an increase in frequency. Another interesting observation from this period was the difference between the larger mobile macrovibrissae and the smaller immobile microvibrissae (Brecht et al., 1997). The macrovibrissae were suggested to mainly locate an object, whereas the microvibrissae were placed upon the object for more fine-scale exploration (Brecht et al., 1997). This suggests functional differences between different whiskers on the pad.

Technological developments, such as high-speed video cameras (Mitchinson et al., 2007), computer vision for whisker tracking (Arkley et al., 2014) and the miniaturisation and wireless possibilities of recording apparatus (Mitchinson et al., 2007), have subsequently enabled whiskers to be observed and characterised during more complex tasks. Orienting behaviours were observed, such as during a head rotation (Towal & Hartmann, 2006).

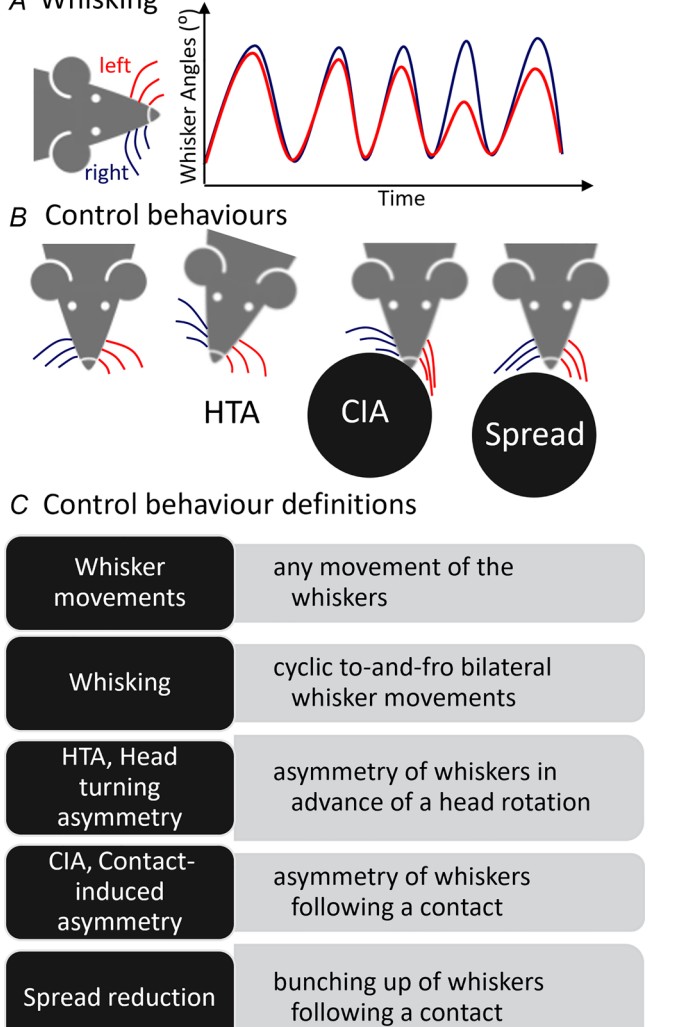

**Figure 1. Summarising whisker movements and control behaviours**
*A*, whisking illustration; left whiskers in red and right whiskers in blue. *B*, control behaviours illustration, including head turning asymmetry (HTA); and two contact related behaviours: contact-induced asymmetry (CIA) and spread reduction (Spread). Control behaviour definitions with abbreviations.

This was termed *head turning asymmetry*, where the whiskers move asymmetrically in advance of a head turn much like a visual saccade (Fig. 1). Whiskers also orient to objects, often touching first with their macrovibrissae and then actively palpating the object with their microvibrissae using head movements (Grant, Sperber et al., 2012), lending support to the observations of Brecht et al. (1997). The microvibrissae area has many densely packed whiskers, and so this behaviour will increase the number of whisker contacts with an object.

Observing and measuring laboratory rodents actively exploring novel objects and surfaces led to the identification of several contact-related whisker control behaviours. These include *contact-induced asymmetry* (Mitchinson et al., 2007), where a rodent reduces the angles of the whiskers ipsilateral to an object to ensure they are not forced hard into the surface, and protracts the whiskers contralateral to the surface more, to increase the number of whisker contacts on the furthest side (Fig. 1); *spread reduction* (Grant et al., 2009), where the spread between whiskers is reduced following a contact, to increase the number of whisker contacts (Fig. 1); and *retraction speed reduction* (Grant et al., 2009), where the speed of the retraction portion of the whisk cycle is reduced so that the whiskers continue to contact the surface for longer durations. Retraction speed reduction has shown that the retraction portion of the whisk can be actively controlled, in agreement with the initial observations of Berg & Kleinfeld (2003).

Social facial touch in rats has been described as producing 'some of the most intense whisker behaviours' and is characterised by long interactions with large movements (Wolfe et al., 2011). Despite this observation, laboratory studies have mainly focussed on describing how rats use their whiskers during object exploration, and much less is known about how they are used in other natural settings, such as during social interactions (Wolfe et al., 2011). Indeed, whisker movements have also been found to guide locomotion, and their positions can change during habituation to an arena (Arkley et al., 2014). This change in the movement and position of whiskers has prompted the suggestion that whisker movements can reveal a 'zone of attention', which can be focussed on to salient spaces in the environment (Arkley et al., 2014; Grant & Arkley, 2016; Mitchinson & Prescott, 2013).

In laboratory whisker studies, rats and mice are the prominent species, and others have been somewhat overlooked. There are some studies on other species of rodentia, including hamsters (who can also whisk) (Wineski, 1983, 1985) and guinea-pigs (who do not whisk, but make more sporadic unilateral whisker movements) (Grant et al., 2017; Jin et al., 2004). The Etruscan shrew has also been studied. Because they are so small, their cortex is very thin and can be imaged using two-photon microscopy (Roth-Alpermann et al., 2010).

Studies with live cricket prey items have revealed the importance of the whiskers during hunting in Etruscan shrews, who use their whiskers to identify spines on the crickets legs to stage precise attacks (Anjum et al., 2006). These findings even led to the development of a bio-inspired prey-pursuit robot, shrewbot (Mitchinson et al., 2012). The marsupial grey short-tailed opossum has also been studied behaviourally (Grant, Haidarliu et al., 2013). They are an interesting species for evolutionary biologists because of their unique phylogenetic position. Grey short-tailed opossums move their whiskers slower than rats and mice (∼7 Hz) and perform contact-induced asymmetry and head-turning asymmetry behaviours (Mitchinson et al., 2011). However, they do not perform spread reduction (Grant, Haidarliu et al., 2013). These findings suggest that both whisking and whisker asymmetry might be common across mammals because they are present in both marsupials and rodents. Such comparative studies can be used to study the evolution of behaviour; for example, Muchlinski et al. (2020) investigated the evolution of whisking, identifying that it probably evolved independently at least seven times.

Both the Etruscan shrew and opossum examples showcase the importance of selecting an appropriate species to address precise scientific questions, including how whiskers are used for hunting, and which whisker movements are present in marsupials. Certain species may be more suitable to a study than laboratory rodents because of their size and ease of imaging (such as in the Etruscan shrew), or their phylogenetic position for studying evolution. However, conducting laboratory studies in different species may be challenging as a result of the required extra space and cost of housing, moving and quarantining protocols, and complexities of maintaining breeding colonies. Therefore, it often might be easier to travel to other captive collections to access different species and conduct studies outside of the usual laboratory.

## Whisker studies in other captive settings

Outside of laboratories, animals can be kept in many other captive settings, including zoos and aquaria, rescue centres and specialised research facilities. Zoos commonly house a variety of mammalian species and have increased the outputs of their research programs in recent decades (Hosey et al., 2019; Kögler et al., 2020; Lina et al., 2020). Indeed, animals in zoo collections can be of high scientific value, above and beyond the roles that they play in education and insurance populations (Kögler et al., 2020). Historically, zoo research was associated with comparative psychology (Hosey et al., 2019) and, although many research areas are now represented, including veterinary, ecology, conservation and physiology (Hosey et al., 2019;

Kögler et al., 2020; Lina et al., 2020), the majority of research is still in vertebrate animal behaviour (Hosey et al., 2019; Lina et al., 2020). The control and analysis of behaviour, especially whisker behaviour, is one area that could specifically support research collaborations between zoos and universities because zoos provide a unique environment for examining the behaviour of species that many academic researchers would not usually have contact with (Fernandez & Timberlake, 2008).

However, working in these types of captive collections may mean that sample numbers are lower than what would usually be expected from lab-based studies, simply because other species are not often kept in large numbers (Grant et al., 2023). Unlike in laboratory animals, access to individuals and species may be limited as a result of their shyness or aggressiveness, facility access, enclosure design and other protocols, such as those for dangerous animals (Grant et al., 2023). These factors can make working in these areas much more challenging than laboratory environments, and this often calls for flexibility in working, as well as developing different study designs for different species and collections. The small, and fast-moving nature of whiskers also makes them particularly challenging to study in more complex environments, such as in a zoo.

Designing simple protocols, inspired from laboratory studies, can often overcome some of these drawbacks. For example, novel object tasks have been applied to small mammals (Grant et al., 2018) using the same high-speed camera and whisker tracking protocols as those developed in the laboratory (Grant, Mitchinson et al., 2012). Using these procedures, whisker movements, but not necessarily rhythmic whisking, have been found to be prevalent across small mammals (Grant et al., 2018). Control behaviours are also common, including the ability to orient and make asymmetric movements, but spread reduction appears to be constrained only to rodents (Grant et al., 2023) (Table 1). Indeed, the muscles required for this behaviour have also only been found in rodents (Grant et al., 2017; Grant, Haidarliu et al., 2013). Novel object tasks have also been applied to more species and collections, using a combination of high-speed and action camera video footage (Grant et al., 2023). Whisker movements have been documented in all species that have been investigated so far, although the degree of contact-related control (contact-induced asymmetry and spread reduction) varies between species (Table 1).

More in-depth studies can also be carried out involving species that can be trained. These studies have primarily targeted pinnipeds, and sometimes otters. These are species that are commonly kept in captivity and are easily trained. Usually, studies are related to object discrimination tasks, such as texture, size or shape, which have been conducted in harbour seal (Grant, Wieskotten et al., 2013), California sea lion (Dehnhardt,

1994; Dehnhardt & Dücker, 1996), South African fur seal (Nakhwa et al., 2024), sea otter (Strobel et al., 2018) and Eurasian otter (Fig. 2*D*) (Nakhwa et al., 2024). These studies have indicated that some species can detect textures to the same resolution as human fingertips, and can do so more quickly (Dehnhardt et al., 1998; Strobel et al., 2018). Other discrimination tasks can also include detecting hydrodynamic vortices, which have been trialled in harbour seals (Krüger et al., 2018) and electromagnetic field detection in cetaceans (Czech-Damal et al., 2012; Hüttner et al., 2022). Unlike in laboratory studies that tend to stimulate whiskers fairly artificially, many harbour seal studies in captivity have made use of more natural stimuli. These include hydrodynamic stimuli that are representative of flatfish breathing (Niesterok et al., 2017), the glide phases from fish swimming (Wieskotten et al., 2010a), the vortex rings from swimming fish (Krüger et al., 2018), moving fish fins (Wieskotten et al., 2010b) and the trails of conspecifics (Dehnhardt et al., 2001; Schulte-Pelkum et al., 2007). However, these studies have only measured discrimination abilities and decision times, rather than quantifying exact whisker positions and movements, which means that identifying specific whisker behaviours associated with these tasks have been overlooked so far.

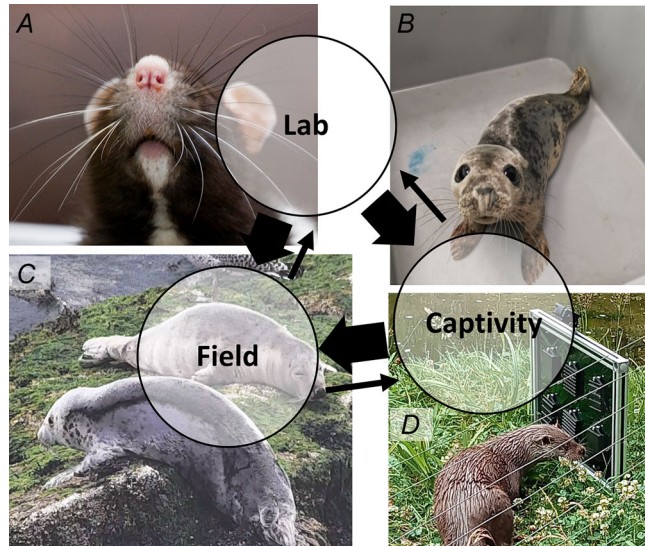

**Figure 2. Research flow of protocols and findings from laboratory studies with studies from other captive institutions (such as zoos, rehabilitation centres and specialist research institutes) and field observations**
Photographs clockwise from top left show a laboratory rat, a seal pup in a rehabilitation centre (photograph from Michal Zatrak), n Eurasian otter in a zoo with enrichment developed in Nakhwa et al. (2024) and grey seals in the field (CCTV footage from Scottish Seabird Centre). Arrows descriptively indicate the flow of research inspiration thus far, with more findings from the lab feeding into captive and field research, and from captivity into field-based research.

**Table 1. Summary of studies that have looked for specific whisker behaviours, including whisker movements, whisking**

| Species | Whisker movements | whisking | HTA | CIA | SR | Other behaviours | Method |
|---|---|---|---|---|---|---|---|
| Laboratory rats[1–5] *Rattus norvegicus* | √ | √ | √ | √ | √ | Social, orienting, locomoting | High-speed camera; automated tracking |
| Laboratory mice[2,6] *Mus musculus* | √ | √ | √ | √ | √ | Locomoting | High-speed camera; automated tracking |
| Grey short-tailed opossum[2] *Monodelphis domestica* | √ | √ | √ | √ | | | High-speed camera; manual/ automated tracking |
| European dormouse[6,7] *Muscardinus avellanarius* | √ | √ | | √ | √ | Locomoting | High-speed camera; automated tracking |
| Etruscan shrew[6] *Suncus etruscus* | √ | √ | | √ | x | Locomoting | High-speed camera[6]; automated tracking |
| Woodmouse[6,7] *Apodemus sylvaticus* | √ | √ | | √ | √ | Locomoting | High-speed camera; automated tracking |
| Yellow-necked[6] mouse *Apodemus flavicollis* | √ | √ | | √ | √ | Locomoting | High-speed camera; automated tracking |
| Harvest mouse[6,7] *Micromys minutus* | √ | √ | | √ | √ | Locomoting | High-speed camera; automated tracking |
| Water shrew[6,7] *Neomys fodiens* | √ | √ | | √ | x | Locomoting | High-speed camera; automated tracking |
| Pygmy shrew[6] *Sorex minutus* | √ | √ | | √ | x | Locomoting | High-speed camera; automated tracking |
| Water vole[6,7] *Arvicola amphibious* | √ | √ | | √ | √ | Locomoting | High-speed camera; automated tracking |
| Bank vole[6] *Myodes glareolus* | √ | √ | | √ | √ | Locomoting | High-speed camera; automated tracking |
| Guinea-pig[6] *Cavia porcellus* | √ | x | | x | x | Locomoting | High-speed camera; automated tracking |
| European hedgehog[7] *Erinaceus europaeus* | | x | | x | x | | High-speed camera; manual tracking |
| Cape porcupine[7] *Hystrix africaeaustralis* | √ | | | √ | x | | Action camera; manual tracking |
| Domestic rabbit[7] *Oryctolagus cuniculus domesticus* | √ | √ | | x | x | | Action camera; manual tracking |
| Domestic ferret[7] *Mustela furo* | √ | √ | | √ | x | | Action camera; manual tracking |
| Weasel[7] *Mustela nivalis* | √ | x | | √ | x | | Action camera; manual tracking |
| European otter[7] *Lutra lutra* | √ | √ | | √ | x | | Action camera; manual tracking |

*(Continued)*

**Table 1. (Continued)**

| Species | Whisker movements | whisking | HTA | CIA | SR | Other behaviours | Method |
|---|---|---|---|---|---|---|---|
| Red fox[7] *Vulpes vulpes* | ✓ | x | | x | x | | Action camera; manual tracking |
| Pacific walrus[8] *Odobenus rosmarus divergens* | ✓ | | ✓ | ✓ | | Feeding | Action camera; manual tracking |
| Harbor seal[8,] *Phoca vitulina* | ✓ | | x | ✓ | x | Feeding, foraging, hydrodynamic | Action camera; manual tracking |
| California sea lion[8] *Zalophus californianus* | ✓ | | x | ✓ | | Feeding, foraging, balancing, hydrodynamic | Action camera; manual tracking |
| Northern elephant seals[9] *Mirounga angustirostris* | ✓ | ✓ | | | | Foraging, hydrodynamic | Head-mounted camera; descriptions |

*Note*: Laboratory studies are indicated by in the top rows in grey, zoos studies in the middle rows in white, and field studies in the bottom rows in grey. Blank cells correspond to behaviours that have not yet been studied in that species. Observations taken from the literature, indicated by the superscript numbers in the species list; 1: Vincent (1912); 2: Mitchinson et al. (2011); 3: Grant et al. (2009); 4: Arkley et al. (2014); 5: Wolfe et al. (2011); 6: Simanaviciute et al. (2020) 6: Grant et al. (2018); 7: Grant et al. (2023); 8: Milne et al. (2020); 9: Adachi et al. (2022).
HTA, head-turning asymmetry; CIA, contact-induced asymmetry; SR, spread reduction.

However, some studies have quantified whisker movements. A size discrimination task in harbour seals showed that, following a whisker contact, seals orient their small rostral whiskers to the object, much like rats with their microvibrissae (Grant, Wieskotten et al., 2013). They then most probably count the number of whiskers contacted to judge size (Grant, Wieskotten et al., 2013), which a quicker and more efficient way than humans, who calculate the distance between the thumb and forefinger (Stevens & Stone, 1959). Indeed. whiskers are often likened to human fingertips. One human touch sensing ability is that of task-specific sensing, where fingertips are moved differently, depending on the task (Lederman & Klatzky, 1987). For example, they stroke textures and squeeze to judge softness. A study in California sea lions showed that they could also make task-specific movements with their whiskers (Milne et al., 2021). The sea lions stroked their whiskers over textures during a texture discrimination task and felt the edges of shapes to judge size during a size discrimination task. This study chose California sea lions to test this idea because this species moved their whiskers more than other pinnipeds (Milne et al., 2020) and had particularly sensitive whiskers (Milne et al., 2022). Indeed, choosing the correct species from which to study specific scientific questions is one justification for conducting comparative work. However, we are often still limited in our species range, with pinnipeds being especially popular for studies using captive collections.

Both novel object and object discrimination tasks are relatively stationary and might not generate the range of movements we might expect from more natural tasks. Looking at balancing or locomotion tasks may offer opportunities to study movement more. Indeed, whisker movements have been shown to guide a dynamic ball-balancing task in California sea lions, with whisker movements preceding compensatory head movements during balancing (Milne & Grant, 2014). Such a task revealed much larger whisker amplitudes (60–120°) (Milne & Grant, 2014) than those observed during a stationary object discrimination task (23–52°) (Milne et al., 2021). Although, they do not move, elephant whiskers are also thought to help guide object balancing (Deiringer et al., 2023). Observing animals locomoting and moving their whiskers is also possible across species and has led to the observation that whiskers scan ahead of forepaw movements in small mammals, probably to guide safe forepaw positioning (Grant et al., 2018). This has been observed on flat and inclined planes (Grant et al., 2018), as well as on branches during climbing in European dormice (Arkley et al., 2017). Measuring dormice whisker movements during gap crossing in a climbing arena revealed larger mean whisker angles (131°) and amplitudes (54°) than when the same animals were observed exploring a simple, flat-floored arena (127° and 38°, respectively) (Arkley et al., 2017), suggesting that more complex experimental settings might cause larger whisker movements.

As well as locomoting, foraging or feeding tasks are also useful to observe whisker movements. However, in captivity, many of the well-studied species, such as pinnipeds and otters, eat fish and are not permitted to

be fed live vertebrate prey. Therefore, feeding enrichment tasks have been designed to encourage more active feeding and foraging. A fish sweeping task was designed for three species of pinniped (Harbour seal, California sea lion and Pacific walrus). Whisker movements and orienting behaviours, including head-turning asymmetry, were observed in all species (Milne et al., 2020). However, mean whisker angles and amplitude values were very similar to those observed in stationary discrimination tasks in harbour seals (angle: 100–150°; amplitude: ∼18°) (Grant, Wieskotten et al., 2013; Milne et al., 2020), and California sea lions even had larger whisker amplitudes in object discrimination tasks than during the fish sweeping tasks (23–52°, compared to 20°) (Milne et al., 2020, 2021). This suggests that discrimination tasks may still cause representative, large whisker movements, despite them being relatively stationary.

An active foraging task was also designed to examine elephant trunk whisker use, by putting food within a box (Deiringer et al., 2023). Moving submarines and moving conspecifics (Wieskotten et al., 2010b; Gläser et al., 2011; Dehnhardt et al., 2014) have also been adopted to study whisker-guided hydrodynamic trail-following in Pinnipeds in a large pool. Whisker movements were not specifically measured in these studies, but it was noted that whiskers were usually protracted during hydrodynamic sensing and did not move much (Wieskotten et al., 2010b). Large, complex captive settings, such as zoo enclosures can offer novel and insightful opportunities to study whisker behaviour during complex locomotion and active foraging tasks, or even during hunting if the prey items are invertebrates. Species-specific, nature-inspired locomotion, foraging or hunting tasks, such as those pioneered by the Dehnhardt laboratory in harbour seals (Schulte-Pelkum et al., 2007; Wieskotten et al., 2010a; Niesterok et al., 2017; Krüger et al., 2018), could easily be simplified and adapted for laboratory settings to pose specific questions about whisker control and sensory processing. These tasks could usefully supplement the classic novel object tasks commonly used in the laboratory.

### Animal studies in the field

One way to study natural hunting and feeding behaviour, especially if the animal is feeding on vertebrate prey, is to look to the wild. Developments of animal-borne tags and video camera technologies (including CCTV and live webcams) (Fig. 2*C*) have enabled the collection of quality video footage from the field with sufficient resolution to view the whiskers (Adachi et al., 2022). Deep-sea cabled video-observatories (Frouin-Mouy et al., 2024) and head-mounted video loggers (Adachi et al., 2022) have been used to capture elephant seals foraging

and feeding underwater. Unlike the observations of seals in captive studies, in the field with live fish, the elephant seals actively protracted and retracted their whiskers, cyclically, similar to that of rodent whisking (Adachi et al., 2022). It was suggested that rhythmic protractions and retractions allow for scanning over larger areas to search for hydrodynamic or tactile stimuli. In addition, active protractions may be energetically costly underwater (engaging a network of muscles), and so retractions might return the whiskers to a rest position to conserve energy. Rhythmic whisker movements have not been documented in any pinniped species before. That it has only been observed during foraging in the wild indicates the importance of studying behaviour in natural settings to gain insights into realistic movements and whisker use. Further field-based studies including more species should give rise to even richer behavioural insights.

As well as video, direct observations can also offer a way to easily capture whisker-use. However, this is fairly manual and time-consuming. Documented whisker-based observations have included whisker rubbing, such as in otariids (Kuhn & Frey, 2012), as well as many social behaviours, especially between mothers and their young in pinnipeds (Evans & Bastian, 1971) and during sexual interactions in manatees (Marshall et al., 1998). Social whisker behaviours are rarely studied in laboratory or other captive settings, but might be particularly important in wild individuals, and should be better understood to gain insights into the natural social behaviours of mammals.

Field and captive studies also do not have to occur in total isolation. Marshall et al. (1998) studied manatees in both the zoo and the wild and described their ability to grasp objects and plants with their whiskers. No differences in whisker movements were observed between the wild and captive individuals. Marshall et al. (1998) went on to qualitatively look for this behaviour across other aquatic species and, although they did not observe it in otariids or phocids, they did find something similar in walruses, as well as full grasping in dugongs. This approach shows how animals in the field may be used to top-up the small sample numbers in zoo studies. Not only this, but also new behaviours observed in the wild can then be searched for in other species, to map behaviours across a phylogeny.

### How can the field inform our captive work?

So far, in the field of whisker science, laboratory studies have mainly been used to inform zoo and field work (Fig. 2); for example, by searching for those originally laboratory-described whisker behaviours, including rhythmic whisker movements, and using novel object tasks or object discrimination tasks to elicit control behaviours, such as contact-induced asymmetry and

spread reduction (Table 1). However, observations from the field can describe rich behavioural repertoires, and help inform our laboratory and zoo-based research. For example, many of the richest whisker interactions in the field are from social interactions, and we do not yet have a protocol that we can use to describe whisker-use during social interactions, especially one that we can apply to many species. Therefore, social whisker behaviours should be studied more across both the field and in other captive settings, including the laboratory. These will probably reveal richer brain signals and novel processing pathways, that will further our understanding of sensing in mammals.

As well as social interactions, foraging episodes, locomotion and environmental interactions are also probably far richer in the field than in a laboratory or other captive environment. Perhaps looking to the field, we can design more realistic foraging tasks or objects for animals to engage with in captivity. Developing realistic, natural objects for animals to interact with in captivity might be especially beneficial for neuroscience studies, aiming to obtain more realistic neuronal responses and behaviours. There could also be wide-reaching welfare applications for producing more active whisker activity and replicating more realistic whisker interactions in captivity. For example, many young seal pups are rescued and rehabilitated in captive settings every year (Zatrak et al., 2023). When pups are very young, they are often kept in isolation and in rather sparse environments (Fig. 2*B*). Developing enriched environments and stimuli to provide more natural sensory experiences will probably help them develop normal behaviours and natural brain development, preparing them better for release.

Both the zoo and field also offer us the opportunity to extend our research away from the usual laboratory animal models. Indeed, with a better understanding of more species, we can select the most fitting species for our scientific questions, whether that be around brain imaging (Roth-Alpermann et al., 2010) or task-specific sensing behaviours (Milne et al., 2021). We can also start to map behaviours over a phylogeny and answer questions on the evolution of behaviour and evolutionary neuroscience. With advancements in camera and biologging technologies, the number of mammalian species that we can access will expand. Indeed, the advent of new wearable sensors makes imaging the whiskers in complex environments more accessible. With technology such as functional near-infrared Spectroscopy (fNIRS), it might even become possible to image the brain of many species in a zoo or field setting (McKnight et al., 2021; Ruesch et al., 2022). The field of whisker science is in the prime position to make use of these new technologies to study animals in their natural environments and develop new protocols to gain deeper insights into the natural sensory behaviours of mammals.

## Current hurdles to overcome

Moving from laboratory studies into the zoo and field does raise a number of challenges that will need to be addressed. The small and fast-moving nature of whiskers makes it hard to image and measure them in complex environments. It is possible to identify only the presence of certain control behaviours (Fig. 1), although this might overlook some important natural behaviours, and does not make use of the rich, quantitative data that can be gathered from tracking whisker movements. Technological developments, especially in cameras, are allowing improvements in imaging, which will probably address this challenge in the future. In addition, zoo and field environments are much less controlled than laboratory environments. There may well be many fewer sample numbers, and scientists need to take a flexible approach when working in these settings; for example, see the recommendations in Grant et al. (2023). As well as technical challenges, zoo and field settings also require additional ethical and risk assessment protocols, such as when working in dangerous environments or with dangerous animals. Nevertheless, zoos, other captive collections and the field can offer whisker researchers a range of species in complex environments, which can allow novel behaviours and interactions to be captured.

So far, the observations from this review suggest that laboratory work is being used to inspire studies in the field and other captive settings. However, these more naturalistic settings are not being used much, if at all, to inspire laboratory tasks. My primary recommendation is that laboratory neuroscientists studying the whisker system would benefit from collaborating with evolutionary biologists and ecologists, aiming to develop more naturalistic, but lab-appropriate protocols. The laboratory will always be needed to provide a controlled environment to probe precise scientific questions, but developing more naturalistic social, foraging and hunting tasks could benefit the field of whisker science by revealing larger whisker movements and more complex whisker control behaviours. Developing more naturalistic tasks will give neuroscientists greater insights into these complex control behaviours, as well as their associated brain processing. It will also be useful for neuroscientists to investigate the suitability of other model species for whisker science, and not limit laboratory studies to rats and mice.

## Conclusions

Because the field of whisker science is already so diverse, many researchers are used to interdisciplinary working. Whisker research is therefore uniquely suited for trialling techniques across the lab, captive and field environments to develop more realistic laboratory protocols, as well as

more standardised field protocols. In the first instance, inspiration from animals in the wild could be used to develop natural sensory stimuli representative of what an animal experiences in the real world and behavioural tasks that enable a broad repertoire of actions. Embedding social and active foraging tasks into suitable laboratory tasks, as well as working across more species, will help us understand whisker behaviour more. This approach has huge implications for the fields of neuroscience, sensory biology and evolutionary biology, as well as for captive mammal health and welfare. Together, these recommendations should give us greater insights into the natural sensory behaviour of mammals.

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

## Additional information

### Competing interests

The authors declare that they have no competing interests.

### Author contributions

R.G was responsible for the conception or design of the work; drafting the work or revising it critically for important intellectual content; approving the final version of the manuscript submitted for publication; and agrees to be accountable for all aspects of the work. All persons designated as authors qualify for authorship, and all those who qualify for authorship are listed.

### Funding

Royal Society (The Royal Society): Robyn A. Grant, APX\R1\211 187

### Acknowledgements

I thank Rasmus Petersen, Miguel Maravall and Riccardo Storchi for organising the Physiological Society meeting 'Breakthroughs in Understanding Natural Behaviour and its Neural Underpinnings', as well as the Physiological Society for funding the meeting. I also thank Charlotte Brassey, Tom Allen and James Gardiner for providing feedback on drafts of the manuscript.

### Keywords

field, laboratory, touch, vibrissae, vibrotactile, zoo

### Supporting information

Additional supporting information can be found online in the Supporting Information section at the end of the HTML view of the article. Supporting information files available:

**Peer Review History**

