## [Peer Review History · The Journal of Physiology]

Can we study whisker movements to gain insights into the natural sensory behaviours of mammals?

Robyn A Grant

DOI: 10.1113/JP288053

Corresponding author(s): Robyn Grant (robyn.grant@mmu.ac.uk)

The following individual(s) involved in review of this submission have agreed to reveal their identity: Michael Brecht (Referee #1)

Review Timeline:

Submission Date:	12-Mar-2025
Editorial Decision:	29-Apr-2025
Revision Received:	07-May-2025
Accepted:	13-May-2025

Senior Editor: Laura Bennet

Reviewing Editor: Ilan Lampl

Transaction Report:

Dear Dr Grant,

Re: JP-SR-2025-288053 "**Can we study whisker movements to gain insights into the natural sensory behaviours of mammals?**" by Robyn A Grant

Thank you for submitting your manuscript to The Journal of Physiology. It has been assessed by a Reviewing Editor and by 2 expert referees and we are pleased to tell you that it is acceptable for publication following satisfactory revision.

ABSTRACT FIGURES: Authors may use The Journal's premium BioRender account to create/redraw their Abstract Figures (and any other suitable schematic figures). Information on how to access this account is here: <https://physoc.onlinelibrary.wiley.com/journal/14697793/biorender-access>.

REVISION CHECKLIST: Upload a full Response to Referees file. To create your 'Response to Referees' copy all the reports, including any comments from the Senior and Reviewing Editors, into a Microsoft Word, or similar, file and respond to each point, using font or background colour to distinguish comments and responses and upload as the required file type.

- 'Potential Cover Art' for consideration as the issue's cover image.
- Appropriate Supporting Information (Video, audio or data set: see https://jp.msubmit.net/cgi-bin/main.plex?form_type=display_requirements#supp).

We look forward to receiving your revised submission.

Yours sincerely,

REFeree COMMENTS

Referee #1:

The review by Grant 'Can we study whisker movements to gain insights into the natural sensory behaviours of mammals?' discusses methods for broadening our understanding of natural whisker behaviors. This is a review paper, thus the question at hand is less if the data present are correct and correctly interpreted, but more so if the topic raised here is important and the message is timely. I'd answer with a resounding yes, we need indeed more and novel ways to study natural whisker behaviors.

Major Points:

1. The study addresses an important issue. While we have rich and compelling lab work, being done on rats and mice, we desperately need more species diversity and more natural whisker behavior work.
2. The review sketches important novel leads as to how study whiskers such as the interplay of work in lab, in captivity and in the zoo.
3. I don't think the review does an optimal job in tracing back, why we have so little work on natural whisker behaviors. Part of the problem is that much of whisker originated from research on cortical barrels. This research tradition is much different from auditory work on bats, which was always driven by the desire to understand echolocation behavior. A second difficulty is that whiskers are hard to see, even in the lab. Such challenges are amplified in field research. A third and major problem is the human don't have whiskers, which results in a lack of intuition for whisker behaviors.
4. The review talks broadly about 'captivity', but more specific comments (and encouragement) for work in zoos would be desirable. Currently we have relatively little whisker (and other scientific) work in zoos. The resource that zoos represent with their hundreds of mammalian species is simply incredible. This being said neither the zoos nor the scientists really embrace this research possibility. If we keep zoo animals, the least we can do is make the animals available for science, this needs to be part of the educational mission of zoos. Scientists in turn need to realize that rat and mouse whisker behaviors are a small part of what is out there.
5. Table 1 underestimates the work that has been done on natural whisker behaviors. For example, in seals many more and more diverse studies have been done, some of which are even made it the references. I think it might make sense to highlight the seal work (hydrodynamic trail following; follicle temperature measurements etc) with pioneers like Dehnhardt as an example of where the field should go.

Referee #2:

Review

This manuscript offers a comprehensive and persuasive overview of whisker science, emphasizing the necessity of integrating laboratory, zoo, and field studies to understand vibrissal function across mammals fully. The authors highlight that while laboratory-based research has been foundational, it often lacks the ecological validity needed to capture the full range of whisker-mediated behaviors and neural processing. By synthesizing findings from diverse settings and species, the review demonstrates how comparative and cross-contextual approaches can reveal behavioral strategies, evolutionary adaptations, and individual variations often overlooked in standard lab models.

The paper strongly advocates for developing more naturalistic experimental paradigms and including a broader taxonomic range, arguing that such efforts will yield more profound insights into sensory processing, behavior, and animal welfare. The review's interdisciplinary scope-connecting neuroscience, behavior, evolution, engineering, and robotics-broadens the conceptual framework of whisker science and encourages collaboration across fields. It also underscores the importance of population diversity and social behaviors, which remain underexplored but are crucial for translating animal research into

real-world contexts.

Despite its strengths, the manuscript is primarily conceptual. It would benefit from including concrete empirical proposals, quantitative comparisons, and more precise definitions of technical terms to enhance accessibility and impact. Additionally, the practical challenges of implementing integrated research, such as logistical, methodological, and ethical considerations, are acknowledged but could be addressed in greater depth.

Overall, this review offers a compelling and timely call to action for the whisker science community, urging researchers to move beyond the confines of laboratory rodents and adopt a more integrated, comparative, and ecologically valid approach. By highlighting the limitations of current laboratory-centric practices and showcasing the potential of cross-setting collaborations-including zoo and field studies-the paper provides a valuable roadmap for future research that could significantly advance our understanding of tactile sensing and behavior. Embracing these broader strategies promises to bridge the gap between controlled laboratory findings and real-world applications, enriching sensory neuroscience's scientific depth and translational impact.

END OF COMMENTS

Thank you so much for the kind and thoughtful comments of the reviewers, I really appreciate their encouragement. Below, I include all the reviewer comments that they suggested for me to address. I have taken care to incorporate every suggestion, which I detail below here and have also added to the manuscript in red text.

Referee #1:

1. I don't think the review does an optimal job in tracing back, why we have so little work on natural whisker behaviors. Part of the problem is that much of whisker originated from research on cortical barrels. This research tradition is much different from auditory work on bats, which was always driven by the desire to understand echolocation behavior. A second difficulty is that whiskers are hard to see, even in the lab. Such challenges are amplified in field research. A third and major problem is the human don't have whiskers, which results in a lack of intuition for whisker behaviors.

Thank you for this comment. I have added to the introduction:

“Many studies focussed on the barrel cortex and tracing the sensory signal from whisker touch to cortex (Petersen, 2019; Staiger & Petersen, 2021). This foundation of whisker science, being firmly embedded in laboratory neuroscience, has impacted the design of research studies. Specifically, simple stimuli, such as air puffs and moveable metal poles, are now standard whisker stimulators (Bosman et al., 2010; Campagner et al., 2018), rather than selecting a stimulus that mimics natural whisker interactions.”

And

“Whiskers are inherently small, and are moved quickly, which makes them hard to see and measure.”

And under the Whisker studies in other captive settings heading:

“The small, and fast-moving nature of whiskers also makes them particularly challenging to study in more complex environments, such as in a zoo.”

And under the Current hurdles to overcome heading:

“Moving laboratory studies more into the zoo and field does raise a number of challenges that will need to be addressed. The small and fast-moving nature of whiskers makes it hard to image and measure them in complex environments. It is possible to identify only the presence of certain control behaviours (Figure 1), although this might overlook some important natural behaviours, and does not make use of the rich, quantitative data that can be gathered from whisker movements. Technological developments, especially in cameras, are allowing improvements in imaging, which will probably address this challenge in the future.”

2. The review talks broadly about 'captive', but more specific comments (and encouragement) for work in zoos would be desirable. Currently we have relatively little whisker (and other scientific) work in zoos. The resource that zoos represent with their hundreds of mammalian species is simply incredible. This being said neither the zoos nor the scientists really embrace this research possibility. If we keep zoo animals, the least we can do is make the animals available for science, this needs to be part of the educational mission of zoos. Scientists in turn need to realize that rat and mouse whisker behaviors are a small part of what is out there.

Thank you for this, I have added to the beginning of the zoo section:

“Zoos commonly house a variety of mammalian species and have increased the outputs of their research programs in recent decades (Hosey et al., 2019; Kögler et al., 2020; Lina et al., 2020). Indeed, animals in zoo collections can be of high scientific value, above and beyond the roles they play in education and insurance populations (Kögler et al., 2020). Historically, zoo research was associated with comparative psychology (Hosey et al., 2019) and while many research areas are now represented, including veterinary, ecology, conservation and physiology (Hosey et al., 2019; Kögler et al., 2020; Lina et al., 2020), the majority of research is still in vertebrate animal behaviour (Hosey et al., 2019; Lina et al., 2020). The control and analysis of behaviour, especially whisker behaviour, is one area that could specifically support research collaborations between zoos and universities, since zoos provide a unique environment for examining the behaviour of species that many academic researchers would not usually have contact with (Fernandez & Timberlake, 2008).”

3. Table 1 underestimates the work that has been done on natural whisker behaviors. For example, in seals many more and more diverse studies have been done, some of which are even made it the references. I think it might make sense to highlight the seal work (hydrodynamic trail following; follicle temperature measurements etc) with pioneers like Dehnhardt as an example of where the field should go.

Thank you. I developed Table 1 to only focus on studies that specifically and primarily study whisker movements and behaviours. This was not clear before, so I have amended the description of the table to: “Summary of studies that have looked for specific whisker behaviours, including whisker movements, whisking, HTA: Head-turning asymmetry, CIA: Contact-induced asymmetry and SR: Spread Reduction.”

I have also added a section on Dehnhardt’s work. This was a really helpful point, thank you.

“Unlike in laboratory studies that tend to stimulate whiskers fairly artificially, many harbour seal studies in captivity have made use of more natural stimuli. These include hydrodynamic stimuli that are representative of flatfish breathing (Niesterok et al., 2017), the glide phases from fish swimming (Wieskotten et al., 2010a), the vortex rings from swimming fish (Krüger et al., 2018), moving fish fins (Wieskotten et al., 2010b) and the trails of conspecifics (Dehnhardt et al., 2001; Schulte-Pelkum et al., 2007). However, these studies have only measured discrimination abilities and decision times, rather than quantifying exact whisker positions and movements, which means that identifying specific whisker behaviours associated with these tasks have been overlooked so far.”

And also:

“Species-specific, nature-inspired locomotion, foraging or hunting tasks, such as those pioneered by the Dehnhardt laboratory in harbour seals (Schulte-Pelkum et al., 2007; Wieskotten et al., 2010a; Niesterok et al., 2017; Krüger et al., 2018), could easily be simplified and adapted for laboratory settings in order to pose specific questions about whisker control and sensory processing.”

Referee #2:

1. Despite its strengths, the manuscript is primarily conceptual. It would benefit from including concrete empirical proposals, quantitative comparisons, and more precise definitions of technical terms to enhance accessibility and impact.

Thank you for this. I do agree, the article is rather conceptual and speculative. I was encouraged to do this through the author guidelines of the article type Symposium review, which states, for example, "Authors are encouraged to be speculative, or even controversial". I have incorporated changes to address this comment though. For example, I have added more empirical evidence in three places in the manuscript:

"Indeed, whisker movements have been shown to guide a dynamic ball-balancing task in California sea lions, with whisker movements preceding compensatory head movements during balancing (Milne & Grant, 2014). Such a task revealed much larger whisker amplitudes (60-120°) (Milne & Grant, 2014) than those observed during a stationary object discrimination task (23-52°) (Milne et al., 2021)."

And:

"Measuring dormice whisker movements during gap crossing in a climbing arena revealed larger mean whisker angles (131°) and amplitudes (54°) than when the same animals were observed exploring a simple, flat-floored arena (127° and 38°, respectively) (Arkley et al., 2017), suggesting that more complex experimental settings might cause larger whisker movements."

And:

"However, mean whisker angles and amplitude values were very similar to those observed in stationary discrimination tasks in harbour seals (angle: 100-150°; amplitude: ~18°; (Milne et al., 2020; Grant et al., 2013b)), and California sea lions even had larger whisker amplitudes in discrimination tasks than during the fish sweeping tasks (23-52°, compared to 20° (Milne et al., 2020, 2021)). This suggests that discrimination tasks may still cause representative, large whisker movements, despite them being relatively stationary.

Following on from these, I add a more empirical proposal:

"The laboratory will always be needed to provide a controlled environment in order to probe precise scientific questions, but developing more naturalistic social, foraging and hunting tasks could benefit the field of whisker science, by revealing larger whisker movements and more complex whisker control behaviours."

In Figure 1, I also have now included a set of abbreviations for the main whisker behaviours too.

2. Additionally, the practical challenges of implementing integrated research, such as logistical, methodological, and ethical considerations, are acknowledged but could be addressed in greater depth.

I have added another paragraph to the section current hurdles to overcome:

"Moving from laboratory studies into the zoo and field does raise a number of challenges that will need to be addressed. The small and fast-moving nature of whiskers makes it hard to image and measure them in complex environments. It is possible to identify only the presence of certain control behaviours (Figure 1), although this might overlook some important natural behaviours, and does not make use of the rich, quantitative data that can

be gathered from tracking whisker movements. Technological developments, especially in cameras, are allowing improvements in imaging, which will probably address this challenge in the future. In addition, zoo and field environments are much less controlled than laboratory environments. There may well be many fewer sample numbers, and scientists need to take a flexible approach when working in these settings (see recommendations in (Grant et al., 2023b)). As well as technical challenges, zoo and field settings also require additional ethical and risk assessment protocols; for example, for working in dangerous environments or with dangerous animals. Nevertheless, zoos, other captive collections and the field can offer whisker researchers a range of species in complex environments, which can allow novel behaviours and interactions to be captured..”

This is in addition to the paragraph in the section on whisker studies in other captive settings:

“However, working in these types of captive collections may mean that sample numbers are lower than what would usually be expected from lab-based studies, simply because other species are not often kept in large numbers (Grant et al., 2023a). Unlike in laboratory animals, access to individuals and species may be limited due to their shyness or aggressiveness, facility access, enclosure design and other protocols, such as those for dangerous animals (Grant et al., 2023a). These factors can make working in these areas much more challenging than laboratory environments, and often calls for flexibility in working, and developing different study designs for different species and collections. The small, and fast-moving nature of whiskers makes them particularly challenging to study in more complex environments, such as in a zoo.”

Dear Dr Grant,

Re: JP-SR-2025-288053R1 "**Can we study whisker movements to gain insights into the natural sensory behaviours of mammals?**" by Robyn A Grant

I am pleased to tell you that your Symposium Review article has been accepted for publication in The Journal of Physiology, subject to any modifications to the text that may be required by the Journal Office to conform to House rules.

NEW POLICY: In order to improve the transparency of its peer review process, The Journal of Physiology publishes online as supporting information the peer review history of all articles accepted for publication. Readers will have access to decision letters, including all Editors' comments and referee reports, for each version of the manuscript and any author responses to peer review comments. Referees can decide whether or not they wish to be named on the peer review history document.

The last Word version of the paper submitted will be used by the Production Editors to prepare your proof. When this is ready, you will receive an email containing a link to Wiley's Online Proofing System. The proof should be checked and corrected as quickly as possible.

All queries at proof stage should be sent to tjp@wiley.com.

The accepted version of the manuscript is the version that will be published online until the copy edited and typeset version is available. Authors should note that it is too late at this point to offer corrections prior to proofing. Major corrections at proof stage, such as changes to figures, will be referred to the Reviewing Editor for approval before they can be incorporated. Only minor changes, such as to style and consistency, should be made a proof stage. Changes that need to be made after proof stage will usually require a formal correction notice.

Are you on Twitter? Once your paper is online, why not share your achievement with your followers. Please tag The Journal (@jphysiol) in any tweets and we will share your accepted paper with our 30,000+ followers!

If you would like to receive our 'Research Roundup', a monthly newsletter highlighting the cutting-edge research published in The Physiological Society's family of journals (The Journal of Physiology, Experimental Physiology and Physiological Reports), please click this link, fill in your name and email address and select 'Research Roundup':
<https://www.physoc.org/journals-and-media/membernews/>

Yours sincerely,

Laura Bennet
Senior Editor
The Journal of Physiology

EDITOR COMMENTS

Reviewing Editor:

Both reviewers agree that the review by Grant calls for expanding whisker research beyond labs, advocating for ecologically valid, species-diverse studies. It highlights current limitations and offers a roadmap for advancing understanding of natural whisker behaviors. The revised manuscript was reviewed by the reviewers, and both support publishing this important paper in its current form.

REFeree COMMENTS

Referee #1:

The authors addressed my concerns. This is an important paper. I support publication.

Referee #2:

I have carefully reviewed the author's response to my comments and suggestions. I am satisfied that all the points I raised have been addressed thoroughly and thoughtfully. The author has provided clear explanations and, where appropriate, made revisions to the manuscript that effectively resolved my concerns. These changes have strengthened the overall clarity and rigor of the work. I have no further objections and support the publication of the revised manuscript in its current form.

* IMPORTANT NOTICE ABOUT OPEN ACCESS *

To assist authors whose funding agencies mandate public access to published research findings sooner than 12 months after publication, The Journal of Physiology allows authors to pay an open access (OA) fee to have their papers made freely available immediately on publication.

You will receive an email from Wiley with details on how to register or log-in to Wiley Authors Services where you will be able to place an OnlineOpen order.

You can check if your funder or institution has a Wiley Open Access Account here: <https://authorservices.wiley.com/author-resources/Journal-Authors/licensing-and-open-access/open-access/author-compliance-tool.html>.

Your article will be made Open Access upon publication, or as soon as payment is received.

If you wish to put your paper on an OA website such as PMC or UKPMC or your institutional repository within 12 months of publication you must pay the open access fee, which covers the cost of publication.

OnlineOpen articles are deposited in PubMed Central (PMC) and PMC mirror sites. Authors of OnlineOpen articles are permitted to post the final, published PDF of their article on a website, institutional repository, or other free public server, immediately on publication.

Note to NIH-funded authors: The Journal of Physiology is published on PMC 12 months after publication, NIH-funded authors DO NOT NEED to pay to publish and DO NOT NEED to post their accepted papers on PMC.